# Mental health issues of children and young people displaced by conflict: A scoping review

ChinenyeOche Otorkpa[1], Oche Joseph Otorkpa[2]*, Ololade Esther Olaniyan[3], Onifade Adefunmilola Adebola[4]

1 College of Health Sciences, Federal University Lokoja, Lokoja, Nigeria, 2 Department of Public Health, Faculty of Health Sciences, National Open University Of Nigeria, Lokoja, Nigeria, 3 Department of Applied Statistics and Decision Analytics, Western Illinois University, Macomb, IL, United States of America, 4 Department of Family Medicine, Federal University Teaching Hospital, Lokoja, Nigeria

* drochejoseph@gmail.com

## Abstract

This research is a scoping review aimed at identifying evidence and studies that address the mental health issues of children and young people forcibly displaced by conflict. It also examines mental health interventions for this population and factors that either favor or worsen their mental health. This issue a major public health issue due to increasing global conflicts that results in the continuous displacement of large populations and the development of new communities where children and young people struggle to re-integrate. This sub-population represents a hidden and at-risk group often not prioritized in planning health interventions for displaced populations. The objective was to identify mental health issues faced by children and young people forcibly displaced by conflict, examine available mental health interventions for this population, and identify factors that favor or worsen their mental health.Four databases (PubMed, ScienceDirect, EBSCO, and ProQuest) were systematically searched for published evidence. Additionally, the King's Fund Library, OpenGrey, DANS data archive, APA website, and WHO were searched for gray literature. After applying strict selection criteria, 27 studies were chosen for a full-text review out of the initially identified 4,548 studies. This review identified depression, post-traumatic stress disorder (PTSD), and anxiety as the major mental health issues in this population. Other issues included somatic disorders, sleep disturbances, nightmares, encopresis, and substance abuse. These disorders, when left untreated, did not diminish over time post-displacement. Mental health was positively influenced by mental health services and religious activities. Negative factors included prior trauma, female sex, poverty, child abuse, parental violence, and separation. Several psychotherapy interventions were found to be effective. In conclusion mental health issues among forcibly displaced children and young people are prevalent and troubling, yet empirical evidence is insufficient. Further research is needed, especially among internally displaced children and youth.

**Data Availability Statement:** All data generated or analyzed during this study are included in this published article and its supplementary information files.

**Funding:** The authors received no specific funding for this work.

**Competing interests:** The authors have declared that no competing interests exist.

# Introduction

## Background

With the turn of the 21$^{st}$ century, and with increasing globalization, there has been an increase in conflicts and attendant human displacement globally. According to the United Nations High Commission for Refugees (UNHCR) records, the number of people forcibly displaced by conflict has almost doubled from 41 million at the end of 2010, to 78.5 million by the end of 2020. Out of this large number of displaced people, about 39% are refugees and asylum seekers who are displaced out of their home country border while larger than half of the number (61%) are people displaced by conflict from one location to another but still within the borders of their country [1].

Forced displacement weakens all aspects of the health care system. It significantly affects universal health coverage and global health security. It causes challenges related to access to health, service delivery, health financing, and monitoring. These challenges all culminate to a public health crisis which lead to poor health outcomes, increased morbidity, and mortality for the displaced population, especially the young people [2]. These young people are unique because they are the most vibrant, energetic and dynamic part of the population and form a foundation for a new generation. This makes it easier for them to be involved in violent actions, risky health behaviors, especially in environments affected by armed conflict. They are disproportionately more affected by health issues, especially concerning sexual and mental health [3].

To address the health problems caused by displacement, several researches have been carried out on the health issues among the population and also among different sub-groups.Literature reviews about these health issues, both for internally displaced people and refugees abound. These report a significant burden of both infectious diseases and NCDs, including mental health disorders among this population with increased mortality risk [4–6]. Several researches have also been carried out about public health issues among sub-population groups who have been internally displaced, especially among the women and younger children.

## Rationale and knowledge gap

While it has been established that there are robust researches on the physical, sexual, and reproductive health effects of displacement on the general population, women, and children; these researches seem to thin down when focusing on the mental health of the children, especially the adolescent children and young people. This maybe because of inadequate mental health facilities among the displaced population [7], and also because mental health issues, except suicide are not obvious and easily recognizable causes of mortality as infectious diseases and maternal and infant issues; thus, humanitarian interventions often overlook this important aspect of health. There is also the misconception that mental health issues tend to 'fade away' with time after relocation and being removed from the violent situation as survival (from violence, emergency and infectious diseases as well as reproductive problems) seems to be more important priority in displaced situations. This may contribute to the less attention paid to mental health issues of the displaced people, especially children and young women. Also, several researches report a plausible increase in PTSD and depression among the displaced population, and little attention is paid to other psychological issues which may be prevalent, thus there are less researches on the other psychopathology of the displaced people [8, 9]. Most information on the mental health issues of this population is based on that provided by practitioner expertise, which though, is a good source of evidence, is not as strong as that based on independent research using robust designs. There is also a paucity of empirical evidence concerning which mental health interventions among this population were effective, much less their comparative effectiveness or their cost effectiveness or scalability [10]. Lastly, it

is also easier to access research on the mental health of the displaced teenage children, than the young people beyond the teenage age. They represent a neglected population as they are neither child nor adult, and are either too old or too young for health interventions targeted at children and adults; this is especially true for young men within this population.

Mental health issues are among the leading causes of illness and disability among adolescents and suicide is the fourth leading cause of mortality among the young people. Thus, failing to address the mental health of this population has consequences that extend well into adulthood and limit the extent these displaced people can truly integrate into the new communities, lead productive lives, and contribute positively to the society [11]. Identification of these mental health issues also serve as a scientific guide for humanitarian aid and programs that address these issues instead of blind assumptions. There is therefore, a need to examine all the primary literature and researches concerning the mental health issues faced by this population. These issues will cut across all psychological issues experienced by both the older children and the youth. This scoping review therefore attempted to fill this gap in knowledge and gather all the available literature concerning the mental health issues of both children and young people in displaced situations.

Therefore, the aim of this study was to identify all researches concerning the mental health needs of displaced children and young people with a view to fill the gap in knowledge and reveal future research needs for policy makers and humanitarian agencies to properly fashion programs that will address the mental health issues of this population.

## Objective

The objectives of this study were:

- To identifythe mental health needs of children and youth displaced by conflict or violence

- To identify factors that affect the mental health of the children and youth displaced by violence.

- To appraise the effectiveness of interventions targeted towards the mental health of children and young people displaced by conflict or violence.

The aim and objectives of this review were to answer the important research question which was framed according to the PICO format [12], as shown below:

P—Population of Interest (Children and Young People)

I–Phenomenon of Interest (Displacement by conflict)

CO- Context/Outcome of Interest (Mental Health needs/Issues)

For clarity and simplicity, the questions were broken down into three components namely:

1. What are the mental health needs of the children and young people displaced by conflict?

2. What factors affect these mental health issues among this forcibly displaced population?

3. How effectively have several health programs and interventions addressed these needs?

## Methods

### Inclusion and exclusion criteria for studies in this scoping review

This review was done using the study flow diagram for scoping review as a guide which is a modification of the PRISMA flow diagram.

In order to harvest as much relevant information as possible for inclusion in this review, all studies (qualitative, quantitative, and mixed studies) on mental health challenges/issues of children and young people forcibly displaced by conflict; as well as all identified studies on interventions and outcomes for mental health on refugee or internally displaced children and young people were considered. Gray studies were also included for consideration in this review.

From these however, some studies were excluded. The criteria used include:

Studies that met the following criteria were excluded. They include only primary research was used. Secondary researches like systematic reviews, scoping reviews, review of literature, editorials, opinions and narratives were excluded.

1. Studies published before 2000 (more than 23 years) were excluded as they may not reflect current realities.

2. Studies on children or young people forcibly displaced by natural disasters wereexcluded because the mental health issues may vary and the effect of conflict on mental health which \is of interest in this review will be lost.

3. Longitudinal studies on children and young people who had experienced war or conflict in the past but were still living in their homes (neither internally displaced nor refugees) were excluded. These studies mostly study mental health effect of war and experiences of war and do not cover an important aspect of forced displacement which is re-settlement and re-integration.

4. Studies on second generation of forcibly displaced children and young people were excluded. In these studies, the studied population were children of refugees and internally displaced people who did not experience the conflicts and displacement firsthand.

5. Studies on children and young people who had successfully resettled and spent more than 10 years in their new or adopted home and are officially citizens of the adopted country or state. These children or young people are too far removed from conflict and its effects in their home countries and may not represent the true picture of mental health problems related to displacement by conflict or violence.

## Data sources

To answer the research questions, five databases were searched extensively and systematically.

Pubmed,Sciencedirect, EBSCO, and Proquestdatabaseswere searched between 16[th] May to 9[th] June 2023 for all information relevant to the topic. Grey literature sites like The Kingfund Library, World cat, AHRQ, Open Gery DANS data archive, APA website, and WHO website were also searched within the same time period.

In each of the databases, a free text search of the topic was first done. Key terms were also selected to locate studies relevant to the research questions. The Boolean search strategy combining keyword and MeSHterms for the main variables was used to improve the yield of the study. The search terms used were as follows: ['Psychiatric disorder' OR 'Psychological issues' OR 'Mental health problems'] for mental health issues; ['children' OR 'adolescents' OR 'youth' OR 'young people'] for children and young people; ['forcibly displaced' OR 'displaced by violence' OR 'displaced by conflict' OR 'refugees' OR 'internally displaced'] for displaced by conflict. The search terms were entered into the databases with an 'AND' term between each of them. The studieswith relevant topics and their abstracts written in English language identified by the above search methods were collated together from the different data bases and uploaded into the Rayaan website.

After upload of the identified relevant topics and their abstracts into the Raayan website, duplicates were easily identified based on a minimum of 96% similarity score and deleted after cross-check and confirmation. The abstracts were then assessed and after exclusion of studies based on the above eligibility criteria, the selected studies were then collated for full text review. Only selected studies with available full texts were reviewed as only the abstracts of some studies were available.

Additionally, the references of selected articles were also searched and relevant articles were selected and screened for inclusion for the study.

## Data charting

After a full text review of the included studies, relevant data that will help answer the research questions on mental health issues of the displaced children and young people; and achieve the set objectives were extracted for analysis. The form was designed by the investigators and cross checked independently by the investigators to ensure all relevant information were extracted. The information extracted included:

1. Author(s)

2. Year of study

3. Type of study

4. Place of study

5. Age range of population (whether younger children, older adolescents or young people)

6. Type of displacement

7. Identified mental health issues

8. Identified associations with mental health issues

9. Intervention (if any)

10. Conclusion/ Key findings

11. Recommendation

The type of displacement was loosely grouped into internal displacement and refugees. Asylum seekers were also grouped as refugees since they had crossed their countries border to seek re-settlement. As with a scoping review, a critical assessment of the sources of evidence was not done, instead, all the relevant data of interest were extracted.

## Results

### Characteristics of sources of evidence

From Fig 1 above, 27 studies fulfilled the selection criteria and were chosen for a full text review. Below is Table 1 which shows a summary of the characteristics of the studies extracted:

### Mental health issues

The extracted mental health issues as reported by the studies were categorized into two broad categories abased on the type of studies. These are the quantitative and qualitative studies.

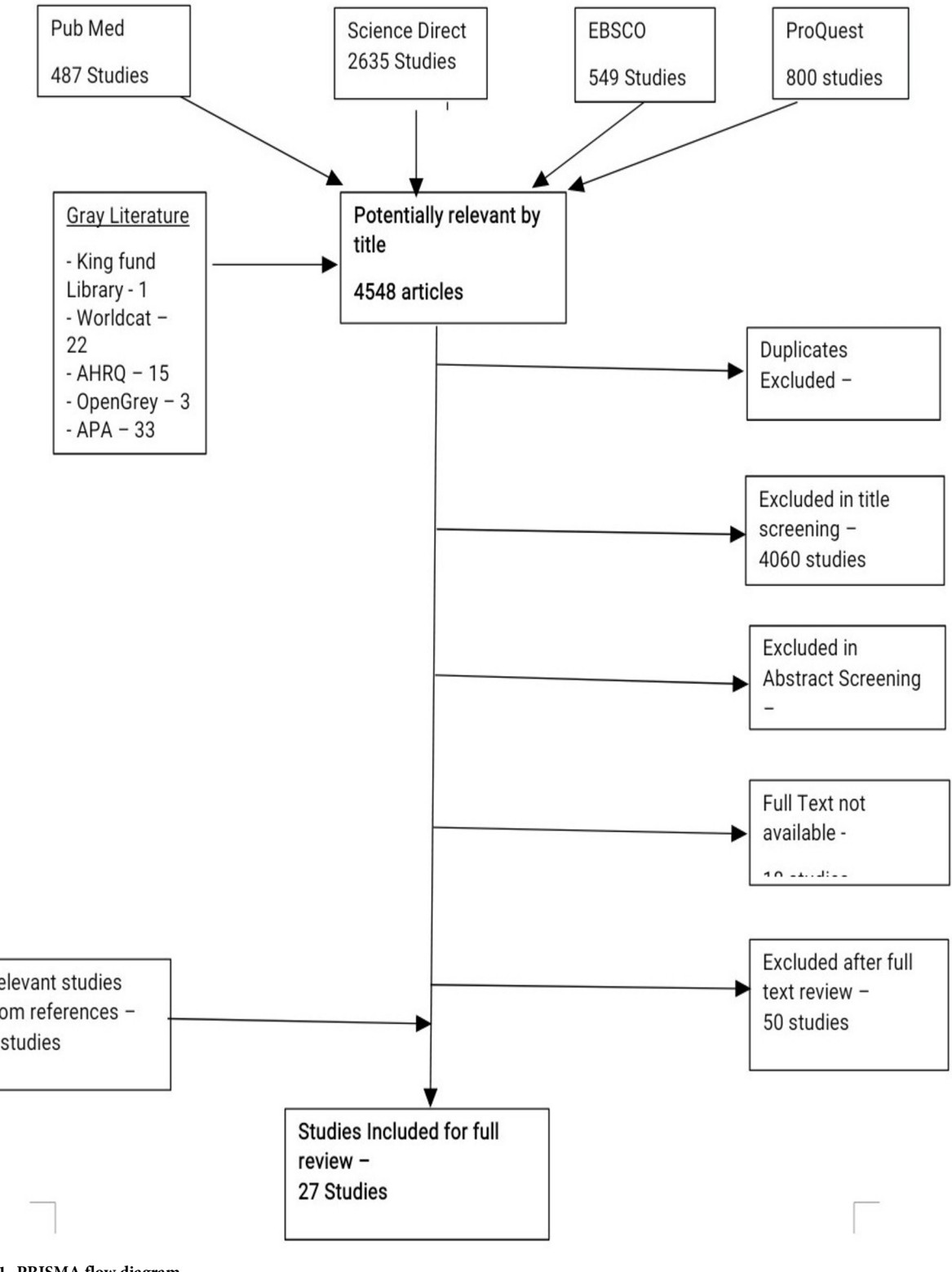

**Fig 1. PRISMA flow diagram.**

**Table 1. Summary of the characteristics of the studies extracted.**

| S/no | Author(s)/Year of study | Place of Study | Population | Age range | Study Design |
|---|---|---|---|---|---|
| 1. | [13] | Australia | Refugees mainly from Iran, Iraq, Afghanistan, and Palestine awaiting resettlement | 11 months– 17 years | Qualitative |
| 2. | [14] | Turkey | Syrian refugees in Temporary accommodation center. | 12–17 years | Cross sectional |
| 3. | [15] | Western Australia | Recently (within 1 year) resettled refugee children | 4–18 years | Cross sectional |
| 4. | [16] | Thailand | Refugee camp | 'Children/Youth' | Qualitative |
| 5. | [17] | Tanzania | Refugee camp | Adolescents | Qualitative |
| 6. | [18] | Nigeria | IDP camp | 13–24 years | Qualitative |
| 7. | [19] | Uganda | IDP camp | 14–17 years | Interventional (Randomized Controlled study) |
| 8. | [20] | Colombia | IDP camps | 12–17 years | Cross sectional |
| 9. | [21] | Turkey | Refugee school children | 8–18 years | Cross sectional |
| 10. | [22] | Uganda | Refugee/ IDP settlement | 16–24 years | Cross Sectional |
| 11. | [23] | Norway | Refugee child care center | Youth below 24 years | Longitudinal study |
| 12. | [24] | Sweden | Refugee Settlement | 13–16 years | Cross Sectional |
| 13. | [25] | Myanmar | IDP camps | Children/adolescents | Qualitative |
| 14. | [26] | Afghanistan | IDP school | 11–18 years | Cross-sectional study |
| 15. | [27] | Syria | IDP settlement | 13–17 years | Interventional (Quasi randomized study) |
| 16. | [28] | Jordan | Resettled refugees | Children | Qualitative |
| 17. | [29] | Jordan | Refugee school | 12–17 years | Cross sectional |
| 18. | [30] | Jordan | Refugee clinic | 2–18 years | Cross sectional |
| 19. | [31] | Sudan | Refugee settlement | Adolescents | Mixed study |
| 20. | [32] | Tanzania | Refugee settlement | 10–14 years | Interventional (Randomized control trial) |
| 21. | [33] | Turkey | Refugee Settlement | 8–17 years | Cross Sectional |
| 22. | [34] | Netherlands | Refugee Clinic | 0–20 years | Cross Sectional |
| 23. | [35] | Turkey | Refugee clinic | 3–17 years | Cross Sectional |
| 24. | [36] | United Kingdom | Refugee mental health service | 3–17 years | Interventional (Non-Randomized control trial) |
| 25. | [37] | Australia | Refugees | 4–18 years | Interventional (Randomized control trial) |
| 26. | [38] | Belgium | Refugees | 14–17 years | Longitudinal study |
| 27. | [39] | Nigeria | Internally Displaced Persons | 14–21 years | Qualitative |

## Quantitative studies

A total of fifteen (15) studies used quantitative methods to describe the mental health issues of the studied population. Out of these, twelve (12) were cross-sectional while the other three (3) were longitudinal studies.

The cross-sectional studies describing the mental health issues were classified into three (3) categories based on the method of diagnosis or description of the issues. These categories are:

- **Category 1:** By the diagnosis according to the Diagnostic Statistical Manual (DSM) IV/V editions. This was mostly done by mental health professionals or people with some training in mental health.

- **Category 2:** By the use of statistical tools. These tools vary and are validated tools that are based on the DSM criteria for diagnosis of mental health issues.

- **Category 3:** By the use of the scores of the Strength and Difficulties Questionnaire. This tool is a standardized, widely used instrument which gives an insight into understanding of adolescents' emotional and behavioral symptoms. It has 25 items that measure four (4) problem mental health areas (emotional, conduct, hyperactivity/inattention, and peer problems) and a fifth area of prosocial behavior. There are 5 questions for each of the sub-scales mentioned and each item can be marked "Not True", "Somewhat True", or "Certainly True", and can be scored with 0, 1 or 2. The total of the first four subscales represent the total difficulty score, while the fifth subscale indicates positive mental health. A set threshold for the total difficulty scale and each subscale is also identified, and scores above these or low scores for the fifth subscale indicate the presence of mental health problems [29].

Mental Health Issues as Diagnosed with the Diagnostic Statistical Manual (DSM) IV/V editions (Category 1) are shown in Table 2 below:

Mental Health Issues identified using validated tools (category 2) are presented in Table 3 below:

- SRQ- Self Report Questionnaire, DSRS–Depression Self Rating Scale, PHQ–Patient Health Questionnaire, HSCL–Hopkins Symptom Checklist, CESDC–Centre for Epidemiologic Studies Depression for Children, CRIES—Child Revised Impact of Events Scale, PCL-C–Post Traumatic Stress Disorder Checklist for Children, SACRED–Screen for Child Anxiety Related Disorder.

Identified Mental Health Issues Based on Pathologically High Strength and Difficulties Questionnaire (SDQ) Scores (Category 3) are shown in Table 4 below:

In describing the mental health issues of the population of interest, three longitudinal studies which followed up a cohort of displaced children and young people, identifying the mental health issues and comparing them among the same population after time were identified. Even

**Table 2. Mental health issues as diagnosed with the Diagnostic Statistical Manual (DSM) IV/V editions.**

| S/no | Mental Health Issues | Study References |
|---|---|---|
| 1 | Depression (Major Depressive Disorder, MDD) | [15, 30, 34, 35]. |
| 2 | Post Traumatic Stress Disorder (PTSD) | [15, 30, 34, 35]. |
| 3 | Generalized Anxiety Disorder (GAD) | [15, 30, 35]. |
| 4 | Specific Learning Disability | [30, 34, 35]. |
| 5 | Separation Anxiety | [15, 30]. |
| 6 | Nocturnal Enuresis | [15, 30]. |
| 7 | Oppositional Deviant Disorder | [30, 34]. |
| 8 | Autism Spectrum Disorder (ASD) | [30, 35]. |
| 9 | Psychotic Disorder | [34, 35]. |
| 10 | Behavioral Disorder | [34, 35]. |
| 11 | Encopresis | [15]. |
| 12 | Sleep Disturbance/ Nightmares | [15]. |
| 13 | Global Developmental Delay | [30]. |
| 14 | Language Disorder | [30]. |
| 15 | Borderline Personality Disorder | [34]. |
| 16 | Somatic Disorder | [34]. |
| 17 | Attention Deficit Hyperactivity Disorder (ADHD) | [35]. |

**Table 3. Mental health issues identified using validated tools.**

| S/No | Mental Health Issue | Reference / Tool Used |
|------|---------------------|------------------------|
| 1. | Depression | 20/*SRQ*,26/ *DSRS*,22/*PHQ*,31/*HSCL-25*,33/ *CESDC.* |
| 2. | PTSD | 20/*PCL-C*,21/ *CRIES*,26/ *CRIES*,24/*CRIES*,33/ *CRIES.* |
| 3. | Anxiety | 20/ *SRQ*,31/ *HSCL-25*,33/ *SACRED* |
| 4. | Psychosis | 20/ *SRQ* |
| 5. | Suicide | 20 |
| 6. | Suicide attempt | 20 |

though these were quantitative studies, they are described differently due to the extra information they present. These are shown in the Table 5 below:

## Qualitative studies

Some of the studies were of the qualitative design. Here, the mental health issues were grouped into themes as described by the study population as shown in Table 6 below.

## Mental health interventions and outcomes among children and youth displaced by conflict

This study also extracted information concerning the mental health interventions among this population which was also a parameter of interest. Below are the results:

A total of five (5) studies reported evidence on different mental health interventions among the population as shownTable 7 below:

## Factors affecting mental health among studied population

The studies further identified factors affecting these mental health issues. These factors are either favorable or unfavorable as shown in Tables 8 and 9 below:

# Discussion

## Keyfindings

This review results reported more evidence on mental health issues of refugee children and youth compared to IDPs. A total of nineteen out of the twenty-seven studies analyzed reported on only refugee children and youth. These studies were also noted to mostly have been carried out in high income countries of Europe and Australia. The other refugee studies carried out were mostly in Asian countries like Turkey and Jordan which are neighbors to Syria, a country that has been engulfed in crisis for about eleven years. The implication of this trend is that the evidence reported in this review is skewed towards the refugees and the resulting data may not

**Table 4. Identified mental health issues based on pathologically high Strength and Difficulties Questionnaire (SDQ) scores.**

| S/No | Mental Health Parameter | References |
|------|--------------------------|------------|
| 1. | Emotional Problems | [21]. |
| 2. | Conduct Problems | [26, 29]. |
| 3. | Hyperkinetic Problems | |
| 4. | Peer Relationship Problems | |
| 5. | Prosocial Behavior | |

**Table 5. Longitudinal / follow-up studies on mental health of children and youth displaced by conflict.**

| S/no | Mental Health Variables/ Measurement Tools | Follow up time | Outcome/ Findings | Reference |
|---|---|---|---|---|
| 1. | • Anxiety / BSI<br>• Depression / BSI<br>• Negative self-concept/ BSI<br>• Hostility/ BSI<br>• GSI<br>*All BSI variables and GSI above pathological threshold* | 3 years | All BSI variables below pathological threshold.<br>GSI significantly reduced. | [14] |
| 2. | • Depression/ HSCL-37A<br>• Anxiety/ HSCL-37A<br>• Externalization / HSCL-37A<br>• Somatization/ CSSI– 8<br>• PTSD/ CPSS | Three time points. viz<br>• Start of study (T1),<br>• 2 years later (T2),<br>• 5 years later (T3) | At T2—No significant reduction in psychological distress and all variables.<br>At T3-<br>Depression—significantly reduced<br>Anxiety, Externalization and Somatization–no significant difference.<br>PTSD—marginally reduced. | [23] |
| 3. | • Anxiety / HSCL-37A<br>• Depression/ HSCL-37A<br>• Internalizing Symptoms/ HSCL-37A<br>• PTSD / RATS | Three time points viz:<br>Start of study(T1)<br>6 months later (T2)<br>18 months later (T3) | T2 and T3 –No change in all variables, all remain high scores. | [38] |

*BSI–Brief Symptom Inventory, GSI–Global Severity Index, HSCL- Hopkins Symptom Checklist, CSSI–Childrens Somatization Inventory Short Form, CPSS–Child PTSD Symptom Scale, RATS–Reactions of Adolescents to Traumatic Stress.

accurately capture the full situation of the IDP children and youth, especially among Africans which has been reported to be the single continent with the highest number of IDPs and make up 44% of global total IDPs [40].

Concerning the mental health issues, this review reports the following mental health problems among the studied population:

**Depression.** The main mental health issue reported in this review is depression. It is interesting to note that all categories of the quantitative studies reported and all reviewed qualitative studies identify depression or depressive-like illness among the population.

**Table 6. Themes of mental health issues among the children and youth displaced by conflict.**

| S/no | Mental Health Theme | References |
|---|---|---|
| 1. | Depression/Depressive symptoms<br>(Sadness, shame, negative world view, deliberate self-harm, loss of interest in daily activities and life, crying spells) | [13, 16–18, 25, 28, 39]. |
| 2. | Anxiety /Restlessness symptoms<br>(Stress, worry, frustration, restlessness, trembling) | [13, 17, 18, 28, 39]. |
| 3. | PTSD / Traumatic Symptoms<br>(Fear, dwelling over thoughts, flashbacks, hypervigilance) | [13, 17, 18, 39]. |
| 4. | Changes in Habit<br>(Disruptive, oppositional, deviant) | [16, 25, 28]. |
| 5. | Sleep Disturbances | [28, 39]. |
| 6. | Alcohol/ Substance abuse | [16, 25]. |
| 7. | Enuresis/ Developmental delays/ Regressions | [13, 28]. |
| 8. | Emotional/ Behavioral Difficulties | [28]. |
| 9. | Somatic Symptoms | [13]. |

**Table 7. Mental health interventions and outcomes among children and youth displaced by conflict.**

| S/No | Mental Health Issue of Interest /Tool | Intervention | Outcome | Reference |
|---|---|---|---|---|
| 1. | Depression / Depressive symptoms | IPT–G culturally adapted to local population | Significant reduction in depressive symptoms | [19] |
| 2. | • PTSD/DTS<br>• Sleep Disorders / SDS<br>• Negative war experience/ KAWES | TRPPT and CPPT | Significant reduction in symptoms and scores on DTS, SDS, and KAWES scores | [27] |
| 3 | • Variables on AYPA subscales:<br>Psychological distress<br>Internalizing symptoms<br>Externalizing symptoms<br>Somatic complaints<br>Prosocial behaviors<br>• PTSD / CPSS<br>• Well-being / SWEMWBS<br>• Functional impairment/ qualitatively derived | 7 weekly group sessions with adolescents, Parents/ Care givers using the EASE Intervention | Significant reduction of scores of:<br>• Psychological distress<br>• Internalizing symptoms<br>• Somatic Symptoms<br>Non- significant improvement in scores of:<br>• Externalizing symptoms<br>• Prosocial behavior<br>• PTSD<br>• Functional Impairment<br>• Well-being | [32] |
| 4. | Variables on SDQ subscale | Early referral to mental health Individualized and Specialized Mental health therapy | Improvement of symptoms in 75% of subjects. Significant improvement in all subscales | [36] |
| 5. | Variables on SDQ subscale: | School mental health service by mental health professionals | At end of school year: Significantly reduced scores in all subscales. Highest in peer problems and hyperactivity. | [37] |

*IPT -G–Group Interpersonal Psychotherapy, DTS—Davidson Trauma Scale, SDS–Sleep Disorders Scale, KAWES—Kubitary-Alsaleh War Experiences Scale, TRPPT—Treatment by Repeating Phrases of Positive Thoughts, CPPT—Cognitive and Positive Psychotherapy, AYPA—African Youth Psychosocial Assessment, CPSS—Child PTSD Symptom Scale, SWEMWBS—Short Warwick-Edinburgh Mental Well-being Scale, EASE—Early Adolescent Skills for Emotions, SDQ–Strength and Difficulties Questionnaire.

**Generalized Anxiety Disorder (GAD)/ anxiety disorders.** In this review, this was either diagnosed as GAD by DSM IV/V criteria, simply as anxiety disorder by various validated screening tools, or feeling of anxiousness/stress/restlessness by qualitative descriptions.

**PTSD.** PTSD is another common mental health issue identified in this review. While the quantitative studies identify it with different validated assessment tools, the qualitative studies describe PTSD as 'nightmares', 'bad thoughts', 'flashbacks' and 'repeat visions'.

**Sleep disturbances/nightmares.** This review also identified sleep disorders and nightmares as a prevalent mental health issue in this population. They often occur as a comorbid disorder to other mental health issues like anxiety and PTSD.

**Changes in behavior (oppositional deviant, disruptive, behavioral disorder).** This finding of changes in behavior as reported in this review ranged from oppositional deviant behaviors to disruptive behaviors. This is commoner in younger children as the early years of development shapes a child's developmental experience.

**Alcohol and substance abuse.** This is another mental health issue among the displaced children and youth identified by this review.

**Table 8. Factors favorable for mental health of studied population.**

| S/no | Factor | References |
|---|---|---|
| 1. | Availability of mental health Services | [27, 30, 32, 35–37]. |
| 2. | Longer time of stay | [14, 36]. |
| 3. | Social Integration | [13]. |

**Table 9. Factors unfavorable for mental health of the studied population.**

| S/no | Factor | References |
|---|---|---|
| 1. | Exposure to traumatic events in home environment (Violence, death of loved ones, etc) | [13, 17, 18, 21–23, 25, 26, 29, 38]. |
| 2. | Female sex | [14, 20–23, 26, 33]. |
| 3. | Poverty | [20, 22, 25, 29, 33]. |
| 4. | Parents mental illness/ Violence | [13, 17, 21, 26]. |
| 5. | Child Abuse/Neglect/Mistreatment | [16, 17, 21]. |
| 6. | Daily Hassles (Harassment, Racism, Stigma, discrimination, Acculturation distress, financial difficulties, restriction of movement) | [23, 31, 38, 25] |
| 7. | Family Separation/ Unaccompanied minors | [15, 24, 29, 34]. |
| 8. | Delay in Education | [13, 20] |
| 9. | Uncertain legal status | [13]. |
| 10. | Forced Detention | [15] |
| 11. | Fighting between caregivers/ Parents | [16]. |
| 12. | Alcohol Use | [16]. |
| 13. | Sexual abuse | [17]. |
| 14. | Physical stress of Fleeing | [18] |
| 15. | Young age | [21] |
| 16. | Older age | [33] |
| 17. | Poor Nutrition | [25] |

**Other psychiatric disorders.** An important result in this review is that just like in the normal population, these displaced children and youth also have a wide range of psychiatric disorders prevalent among children and youth apart from Depression, PTSD, and anxiety. They include: Enuresis/encopresis/developmental delays, Somatic disorders, Psychosis, ASD, ADHD, language disorder, and specific learning disorder.

This review also reported three interesting studies. These studies [14, 23, 38] were longitudinal studies that identified the mental health issues among this population and followed them up over time to ascertain if they will 'just go away' and if time had any effect on these issues. The above studies provide empirical evidence that these mental health issues do not just fade away with time among this population without concerted efforts, instead, it may get worse without intervention.

Concerning mental health interventions, three interventions [19, 27, 32] employed some form of psychotherapy technique by trained researchers not limited to mental health professionals. Even though they involved psychotherapy, they were mostly easy to perform. [19] employed group interpersonal psychotherapy (IPT-G) which was culturally adapted to the setting on depressed children and adolescents, while [27] taught children daily repetition of positive phrases in response to stressors over a time period. [32] simply used the WHO developed tool- the EASE intervention which consisted of group sessions of about 90 mins of evidence-based cognitive behavioral strategies for the adolescents as well as two-hour group sessions for the caregivers.

These interventions were cheap and reproducible, and were also reported to be effective in improving the mental health of the studied populations. The other two interventions involved community-based mental health service [36], and school-based mental health service [37]. The thrust of these two interventions were on early referral to easily available mental health service by mental health professionals. They provide empirical evidence that provision of mental health services as part of primary care among displaced population is effective [36].

## Strengths and limitations

The scoping review is a relatively new research method which draws its strength from its thoroughness and transparency, making the results reproducible. By identifying all researches on the topic of mental health problems of children and youth displaced by conflict, it offers a broad information base on the topic and a tool for effectively mapping all available literature on the subject. Its flexibility in inclusion of both published and gray or unpublished literature on mental health issues of displaced children and youth, as well as inclusion of all evidence about the topic irrespective of study methodology, unlike the systematic review ensured it yielded more evidence. Another strength of the scoping review as was noted in this research is the ability to combine qualitative and quantitative evidence. This makes its approach less rigid than the traditional systematic review method. This research work like a typical scoping review drew attention to the state of research activity on mental health issues of children and youth displaced by conflict rather than evaluating the quality of the evidence presented [41].

The main limitation of this scoping review was the difficulty in accessing major databases like SCOPUS and also the non-availability of some chosen full papers. To solve this, alternate databases had to be used and some papers had to be excluded since their full texts were not available. This may serve as a limitation to the results of this study.

Also, an important research gap was noticed in the course of this review. There are very scanty studies on child and youth mental health among the IDPs. The few available ones are mostly qualitative and involved a small sample size. While a qualitative study is an invaluable and recognized source of evidence, especially for deeper exploration of observations, quantitative studies also contain valuable information and provides a baseline for statistical calculations and mathematical measure of relationships.

## Comparison with similar researches

The spread of the studies reported in this review in mostly high-income countries of Europe and Australiamaybe because European countries and Australia have a well-developed and harmonized policy for refugee migration and resettlement [24, 13]. Australia for example, has a detention center from where the applications for resettlement and asylum are reviewed [13]. These make it easier to report trends among the children and young people in these countries. The other refugee studies carried out were mostly in Asian countries like Turkey and Jordan which are neighbors to Syria. Syrian refugees make up the vast worldwide majority of refugees and there are about 5.6 million Syrian refuges worldwide, especially in Turkey [42]. The fact therefore, that most of the studies on refugees were among Syrians is not a very surprising finding.

It is also not surprising that fewer studies on mental health of children and youth IDPs were reported. In this review, only eight of such studies were reported. These studies were mostly in low- and middle-income countries of Africa and Asia like Nigeria, Uganda, and Myanmar which have been engulfed in longstanding internal crisis and insurgency, hence the large number of IDPs in these areas[18, 19, 25]. Evidence may also not be robust on these issues in this population because in these countries with weak health systems, emphasis is often on emergency health services and communicable diseases, and also on women and children health, rather than on non-communicable diseases like mental health whose impact may not be clearly apparent in the early stages [43].

## Explanations of findings

The mental health issues identified in this study among children and youth displaced by conflict reveals a significant burden, with depression. This is not surprising as the World Health

Organization (WHO) has reported that displaced populations face a higher prevalence of common mental disorders, including depression, compared to the general population [7]. A meta-analysis further supports this finding, showing a pooled prevalence estimate of 26.4% for depression and elevated risk of suicide among the displaced [44].

Generalized Anxiety Disorder (GAD) and anxiety disorders are also prevalent among displaced individuals, as supported by WHO's recognition of anxiety as a common mental health issue among displaced persons [7]. Furthermore, Bürgin et al.'s study reports a pooled prevalence of anxiety disorders at 15.8% among displaced children [45]. PTSD is another common mental health issue identified in this review, with reported prevalence ranging from 22.7% to 63.5% in displaced children [45, 46].

The review also outlines behavioral changes, including oppositional deviant and disruptive behavior, along with sleep disorders, aligning with evidence that early exposure to conflict-related stress negatively affects the psychological development of children [47]. The occurrence of Alcohol and Substance abuse among this population may be because adolescence is a recognized risk factor for alcohol and substance abuse as well as the fact that the stressors associated with displacement and coping with conflict can lead to substance abuse among the adolescents and youth. There is also empirical evidence that alcohol and substance abuse easily occur among displaced populations [48, 49].

Three longitudinal studies [14, 23, 38] reported that mental health issues may not naturally resolve over time. While one study reported significant improvement attributed to changes in temporary accommodation centers, others suggested a limited reduction or persistence of mental health symptoms. This challenges the general belief that some mental health issues may naturally diminish with time, emphasizing the need for intervention [50, 51].

The success of the mental health interventions as reported [19, 27, 32] gives hope on the issue of mental health of these displaced children. These interventions affirm the established fact that psychotherapy remains a major treatment for mental health issues [52]. The other two interventions [36, 37] which involved community and school -based mental health services were based mostly on early referral to easily available mental health service by mental health professionals. They provide empirical evidence that provision of mental health services as part of primary care among displaced population is effective [36].

This review implies that the displaced population of children and youth experience a wide variety of psychopathology which are complicated by their special situation of forced displacement and uncertain future. Beyond immediate survival concerns, these individuals face challenges in integration and reintegration, impacting their quality of life into adulthood. The study advocates for increased attention to mental health issues in displaced populations, urging governments and organizations to view survivors of conflicts as a 'population at risk.' It calls for the inclusion of mental health services in interventions, policies, and planning for temporary accommodations. The identified interventions with favorable outcomes are presented as feasible and cost-effective, with the potential to improve the well-being and productivity of the affected population.

## Conclusion

In conclusion, this scoping review sheds light on the often overlooked mental health issues affecting children and youth displaced by conflict. The evidence presented emphasizes the urgent need for sustained interventions that incorporates mental health services, to address the long-term well-being of this vulnerable population beyond immediate survival concerns.

## Supporting information

**S1 Checklist. PRISMA-ScR checklist.**
(DOCX)

## Author Contributions

**Conceptualization:** ChinenyeOche Otorkpa.

**Methodology:** ChinenyeOche Otorkpa, Oche Joseph Otorkpa.

**Project administration:** Oche Joseph Otorkpa, Ololade Esther Olaniyan.

**Supervision:** ChinenyeOche Otorkpa.

**Validation:** ChinenyeOche Otorkpa, Oche Joseph Otorkpa.

**Visualization:** ChinenyeOche Otorkpa.

**Writing – original draft:** ChinenyeOche Otorkpa.

**Writing – review & editing:** ChinenyeOche Otorkpa, Oche Joseph Otorkpa, Ololade Esther Olaniyan, Onifade Adefunmilola Adebola.

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
