## [Decision Letter · Decision Letter 0]

19 Sep 2024

PMEN-D-24-00210

Mental Health Issues of Children and Young People Displaced by Conflict: a scoping review

PLOS Mental Health

Dear Dr. OTORKPA,

Thank you for submitting your manuscript to PLOS Mental Health and we apologise for the severe delay in reaching a decision. After careful consideration, we feel that it has merit but does not fully meet PLOS Mental Health’s publication criteria as it currently stands. Therefore, we invite you to submit a revised version of the manuscript that addresses the points raised during the review process.

Please ensure that you fully address all of the comments raised by the reviewers and in particular, give careful consideration to the extensive points raised by reviewer 3. We will not be able to send this back to reviewers without reviewer 3's comments being addressed.

We look forward to receiving your revised manuscript.

Kind regards,

Karli Montague-Cardoso

Executive Editor

PLOS Mental Health

Journal Requirements:

1. Please provide a/amend your detailed Financial Disclosure statement. This is published with the article. It must therefore be completed in full sentences and contain the exact wording you wish to be published.

**Please only choose the relevant sentences from below**

1. Please clarify all sources of funding (financial or material support) for your study. List the grants (with grant number) or organizations (with url) that supported your study, including funding received from your institution. 

2. State the initials, alongside each funding source, of each author to receive each grant.

3. State what role the funders took in the study. If the funders had no role in your study, please state: “The funders had no role in study design, data collection and analysis, decision to publish, or preparation of the manuscript.”

4. If any authors received a salary from any of your funders, please state which authors and which funders.

2. We note that your Data Availability Statement is currently as follows: "All data generated or analyzed during this study are included in this published article and its supplementary information files."

3. Please provide separate figure files in .tif or .eps format.

https://journals.plos.org/mentalhealth/s/figures 

https://journals.plos.org/mentalhealth/s/figures#loc-file-requirements 

4. We have noticed that you have Table in the manuscript file but there are no corresponding labels in the manuscript. Please amend your manuscript to include this table, noting that tables should not be uploaded as individual files.

5. We have noticed that you have uploaded Supporting Information files, but you have not included a list of legends. Please add a full list of legends for your Supporting Information files after the references list. 

Additional Editor Comments (if provided):

Reviewers' comments:

Reviewer's Responses to Questions

**Comments to the Author**

1. Does this manuscript meet PLOS Mental Health’s publication criteria? Is the manuscript technically sound, and do the data support the conclusions? The manuscript must describe methodologically and ethically rigorous research with conclusions that are appropriately drawn based on the data presented.

Reviewer #1: Yes

Reviewer #2: Yes

Reviewer #3: No

2. Has the statistical analysis been performed appropriately and rigorously?

Reviewer #1: Yes

Reviewer #2: N/A

Reviewer #3: I don't know

3. Have the authors made all data underlying the findings in their manuscript fully available (please refer to the Data Availability Statement at the start of the manuscript PDF file)?

Reviewer #1: Yes

Reviewer #2: Yes

Reviewer #3: Yes

4. Is the manuscript presented in an intelligible fashion and written in standard English?

Reviewer #1: Yes

Reviewer #2: Yes

Reviewer #3: No

5. Review Comments to the Author

Reviewer #1: I have posted my comments on the draft and attached it.

Kindly fix the sentence framing in the introduction part, kindly explain Ryan website as well as include adjustment disorder if you have seen in literature.

Reviewer #2: Data source - explicitly described in the paper

well written paper- structured and easy to read;

the scope of this study was strictly defined as a scoping review which confined the parameters to strict inclusion/exclusion criteria. The papers cited/ references suitable relevant

additional comments to editor:

I would suggest that the authors use the UNHCR data on forced displacement closest to the time of study or to cited reference which is 2023 - should be at least 2022 here: https://www.unhcr.org/sites/default/files/2023-06/global-trends-report-2022.pdf

add some additional context relevant to this study- children make up 40% of all forcibly displace people

was there a reason why gender was not included in the Data Charting? linked the words "female sex" used in the text- need some clarification as - there is evidence of rapes, gender based violence, other sexual exploitations of displaced people including very young girls and sometimes boys

slight adjustment of paper - recommend publication

comments to authors; well presented paper, structured and focus

would encourage further hypothesis on this subject that is a global challenge

Reviewer #3: Thank you for the opportunity to review this article on an important topic of mental health and mental health services of children and adolescents impacted by forced displacement.

Unfortunately, and based on my careful review of the abstract, background and methods section, I am not able to recommend this article for publication. There are significant issues with the paper that require attention.

The background section is not well cited and does not adequately engage with the empirical literature on this topic which is quite extensive. More detail and specificity on what is known and gaps that remain are needed to enhance the quality of the background section. More importantly, there are a number of different reviews of mental health of children impacted by war, conflict and forced displacement which have already been conducted and published. There needs to be a thorough discussion of this existing literature and clear distinction of what is known from these reviews and what gaps remains which this review seeks to address.

Overall, the quality of writing needs improvement with overall copyediting needed.

The methods section overall is lacks key details (some of which I have elaborated below) related to guidelines that were used to conduct the study, search terms and how they were chosen, clarification of inclusion criteria and how selected records were reviewed and appraised. Based on the description, it is not clear if the data extraction tool was develop a priori (a requirement for review) or post-review.

There are currently no details on the analytic methods employed to analyze the data. I have included some additional detailed comments below to elaborate on my summary here and which I hope may be helpful to the authors as they continue to work on their manuscript.

The background literature section is not detailed enough in terms of summarizing the state of the existing literature on forcibly displaced populations, what key findings have been identified? With what populations? Etc. The background literature is also not well-cited in a number of places. There have been many different reviews of the literature that need to be referenced and within which, the current review should be situated. The reviews that appear to be cited seem specific to the African context.

Overall, I find the background section to be lacking rigor in terms of really teasing apart the key issues and needing much more specificity.

Much more engagement with the existing literature on mental health concerns, interventions and services of children and adolescents impacted by forced displacement is needed, particularly engagement with existing reviews which have been conducted. Given that numerous reviews on this topic have already been conducted, what gap is this scoping review seeking to address? The particular rationale for this paper is not clear in the existing manuscript.

Methods

Authors need to specify which specific review guidelines were followed to conduct this review. Please provide a working definition for the scoping review undertaken.

Included a detailed list of inclusion criteria.

Explain more detail about how the relevant databases were selected. Please include details on the study protocol that was developed and if the protocol was registered or is accessible.

Indicate when the searches were conducted and by whom.

Include an appendix with full search and mesh terms for the various databases. Some of the keywords appear problematic as they are included as plurals which may have limited the authors potential to identify relevant studies.

More detail on the title/abstract review and full text review are needed. Who conducted those reviews, how were conclusions reached, why were articles excluded?

There are no details on the methods of analysis used to examine and synthesize the data.

6. PLOS authors have the option to publish the peer review history of their article (what does this mean?). If published, this will include your full peer review and any attached files.

**Do you want your identity to be public for this peer review?** For information about this choice, including consent withdrawal, please see our Privacy Policy.

Reviewer #1: No

Reviewer #2: **Yes: **Albert Persaud

Reviewer #3: No

---

## [Editor Report · Decision Letter 1]

15 Oct 2024

Mental Health Issues of Children and Young People Displaced by Conflict: a scoping review

PMEN-D-24-00210R1

Dear Dr OTORKPA,

We are pleased to inform you that your manuscript 'Mental Health Issues of Children and Young People Displaced by Conflict: a scoping review' has been provisionally accepted for publication in PLOS Mental Health.

Best regards,

Diego Gomez Baya

Academic Editor

PLOS Mental Health